# A comprehensive crop suitability assessment under modern irrigation system in arid croplands

**Ahmed S. Abuzaid[1], Hassan H. Abbas[1], Mostafa A. Mostafa[1], Yousif K. El Ghonamy[2], Nazih Y. Rebouh[3], Mohamed S. Shokr [4]***

**1** Soils and Water Department, Faculty of Agriculture, Benha University, Banha, Egypt, **2** Soils, Water, and Environment Research Institute (SWERI), Agricultural Research Center (ARC), Giza, Egypt, **3** Department of Environmental Management, Institute of Environmental Engineering, People's Friendship University of Russia (RUDN University), Moscow, Russia, **4** Soil and Water Department, Faculty of Agriculture, Tanta University, Tanta, Egypt

* mohamed_shokr@agr.tanta.edu.eg

## Abstract

Agricultural suitability analysis using traditional methods is still arguable due to the uncertainty and subjectivity resulting from manual evaluations. The current work provides a novel framework for integrating the analytical hierarchy process (AHP) with fuzzy logic under the geographic information system (GIS) platform to generate suitability maps for cultivating wheat, broad bean, and maize under center pivot irrigation systems. The research was executed in an arid region (30229 ha) in the western Nile Delta fringes, Egypt. Meteorological data, digital elevation model, and samples collected from seventy soil profiles and fourteen artesian wells were analyzed to characterize local climate conditions, landscape characteristics, and irrigation water quality. The main and sub-criteria were prone to AHP to specify the relative importance (weight) of each factor. Using GIS tools, raster layers were created, assigned scores (fuzzy membership functions) according to crop requirements, and complied in accordance with the weighted sum algorithm to produce final crop suitability maps. Results revealed that climate conditions were highly (S1) and moderately (S2) suitable for winter crops (wheat and broad bean) but marginally suitable (S3) for summer crop (maize). Soil salinity, sodicity, and depth were the most important determinants of landscape suitability. Accordingly, the land resources in the studied region were suitable (S1, S2, and S3) for the selected crops; nevertheless, 193 and 275 ha were currently not suitable (N1) for broad bean and maize, respectively. Potential salinity and specific ion (sodium and chloride) toxicity hazards were the main constraints for groundwater irrigation. The center pivot irrigation would meet wheat and maize requirements but adversely affect broad bean yield. Overall, groundwater quality contributed to 46% of site suitability for crop production followed by landscape factors that contributed to 42% and climate conditions that accounted for 13%. The final suitability maps affirmed high priority for wheat cultivation in the studied region since the

**Data availability statement:** All relevant data are within the manuscript.

**Funding:** This publication has been supported by the RUDN University Scientific Projects Grant System, project No. <202786-2-000>».

**Competing interests:** The authors have declared that no competing interests exist.

S1 and S2 classes encompassed 90 and 10%, respectively. Moreover, maize ranked as the second suitable crop with 55, 42, and 35 of the total area fitting the S1, S2, and S3 classes, respectively. The third place was due to broad bean with S2 and S3 classes representing 53 and 47% of the total area, respectively. Our study can offer a replicable framework to integrate AHP with GIS-fuzzy logic for sustainable food crop production in drylands.

## 1. Introduction

Water scarcity, climate change, and rapid population growth are major threats to mankind in the 21st century [1]. These problems render ensuring food security and safety without deteriorating the natural ecosystem's resources a major challenge, especially for developing nations [2]. To address this issue, the expansion of irrigated croplands is essential, particularly in arid regions [3]. Arid croplands represent about 50% of the world's cultivated area owing to appropriate temperatures year-round that sustain plant growth and expanding irrigation and transport [4]. Despite being implemented for only 20% of croplands, irrigation supports about 40% of food production around the globe [5] and serves as an effective practice to cope with climate change, mitigate drought stress, and improve crop yield and quality [6]. Moreover, irrigated croplands are usually more productive compared with rainfed systems [7]. Yet, the increased water demand advocates an urgent need to shift traditional irrigation systems to water-saving techniques to enhance water use efficiency.

Modern agricultural water management entails optimized exploitations of available water resources via advanced technologies [8]. Center pivot irrigation (CPI) can diminish the gap between water supply and demand in dryland agriculture. This system is widely adopted all over the world, especially in regions depending entirely on groundwater irrigation [9–11]. The CPI is highly preferred in large-scale projects due to low water loss, high irrigation uniformity, low operational and maintenance costs, and the potentiality of irrigating uneven terrain [3,5]. It also causes significant reductions in air temperature, vapor pressure deficit, and vegetation canopy temperature, thus alleviating heat pressure and enhancing crop yield [8]. It has proven effective in stabilizing and restoring light-textured soils, shifting them to arable lands [12]. In Egypt, CPI is practiced on about 207,000 hectares in the newly reclaimed desert areas [7]. Thus, evaluating site suitability for crop production under this modern irrigation system is essential for sustainable land use planning.

Landscape and water resources are pillars for irrigated cropping systems [13,14]. Thus, deep spatial knowledge of terrain, soil, and irrigation water is essential to match their properties with crop requirements [15,16]. This entails handling massive and complex datasets, posing major challenges for decision-making, especially in regional-scale planning [17,18]. Among the main constraints, simple or supervised ratings assigned to environmental parameters, which are classically delineated using distinctive and consistent intervals [19]. Thus, advanced scoring techniques able to address ambiguity and simplify complex interactions among various factors are

crucial to enhance the accuracy of suitability analysis [20]. In this context, fuzzy logic is extensively utilized in expert systems to handle uncertainties and imprecise information [21,22]. Fuzzy models offer a flexible approach to integrate human knowledge with fuzzy membership functions, providing practical solutions [19,23,24]. The fuzzy sets integrated with GIS (geographic information system) have proven to be an efficient approach in crop suitability studies such as chickpea in a semi-arid region of Iran [25] and wheat in an arid region of Egypt [26]. Therefore, fuzzy-based crop suitability analysis can augment the reality of natural resources assessment.

In fact, utilizing fuzzy logic solely for the purpose of modeling crop suitability is insufficient [27,28]. This is because not all factors exert the same level of importance, and the relevance of each element to site suitability varies [29]. In this context, the analytical hierarchy process (AHP) introduced by Saaty [30] is extensively employed to make multi-criteria decisions on site suitability for a certain use [19,31,32]. The AHP offers a simple, dynamic, and adaptable technique to weight each criterion based on how much it contributes to site performance [31]. It proves beneficial in situations where defining specific relationships among numerous criteria is challenging [33]. Furthermore, this approach facilitates the determination of weights for both individual and grouped criteria, making it applicable across various dimensions of land evaluation research [34]. In semi-arid ecosystems, a hybrid AHP-fuzzy logic approach under the GIS platform has proven success in reliable spatial analysis of quantifying land suitability for maize in southern India [35] and rapeseed in the upper Tigris basin of Turkiye [19]. Therefore, to achieve self-sufficiency in food crops, this integrated framework should be implemented under locally dominant conditions in arid croplands.

Cereals and legume grains serve as stable food all over the world, especially for low-income nations, and are the main sources of nutrients and dietary energy [36,37]. Cereal and legume crops are predominantly cultivated in drylands in order to support food supply for most residents, provide feed requirements for livestock production, and enhance exports [38]. Furthermore, these crops yield high amounts of straw, which are recycled to produce compost and biochar [39] and generate biofuels [40]. This, in turn, can improve agroecosystem functions, enhance environmental safety, and mitigate climate change effects through decreasing greenhouse gas emissions [41] and increasing carbon sequestration in agricultural lands [42]. Therefore, cereal and legume-cultivated areas in the dryland agroecosystems have increased during the last few decades owing to their social, economic, and ecological benefits [38]. However, these crops are more affected by adverse impacts of accelerated climate change [43]. Consequently, an accurate site suitability assessment is a prerequisite for ensuring the sustainable production of these strategic crops.

In Egypt, wheat (*Triticum aestivum* L.), broad bean (*Vicia faba*), and maize (*Zea mays* L.) serve as the main components of human diets. According to official statistics [44], the country faces production gaps estimated by 53, 71, and 27%, respectively. Thus, to achieve self-sufficiency in these crops, expanding the cultivated areas in the newly developed desert lands depending on CPI systems should be considered. During the last few decades, giant land reclamation projects have been launched in the Egyptian deserts [2], particularly in the western fringes of the Nile Delta [45,46]. The agricultural expansion in this promising region helped in solving social problems such as unemployment and housing shortage. Nearly, 250,000 residents and their families have settled in this region, including beneficiaries like unemployed graduates, previous tenants, and small farmers [47]. The western Nile Delta region has attracted great attentions due to its distinct geographic location, easy accessibility, gently undulating topography, and mild climate conditions [48]. Additionally, the region is characterized by adequate groundwater resources in terms of quantity and quality due to the presence of four water-bearing strata, involving Oligocene, Miocene, Pliocene, and Quaternary aquifers [49]. However, intensive agricultural and industrial activities in the western Nile Delta region have raised the stress on available land and groundwater resources [48]. Therefore, accurate assessment and monitoring of local natural resources is essential to ensure sustainable food crop production.

Using a hybrid AHP-GIS approach, Abuzaid and El-Husseiny (15) developed spatial site suitability models for cultivating wheat, maize, and broad bean under micro-irrigation systems in the west Nile Delta region. The study categorized suitability classes using simple scores ranging from 5 to 1 to quantify the optimal and worst fitness conditions, respectively.

Yet, deep geographic knowledge about the contribution of each environmental factor to crop performance is quite limited. For this motivation, the present work was conducted to offer an innovative framework via integrating AHP with fuzzy logic under the GIS platform for modeling and mapping site suitability for the potential production of wheat, maize, and broad bean under the CPI system. The model was implemented in a newly reclaimed area located in the western Nile Delta fringes, Egypt to assist decision-making on sustainable food crop production.

## 2. Materials and methods

### 2.1. Site description

The investigated area occupies 302.29 km² (30229 hectares) in the western fringes of the Nile Delta, Egypt. The geographic location is in the UTM zone 36 between latitudes 30° 11′ 20″ to 30° 26′ 00″N and 29° 52′ 44″ to 30° 14′ 09″E (Fig 1). The study does not involve the collection of plant, animal, or other materials from a natural setting. Furthermore, since the study was carried out on publicly accessible property and excluded interactions with controlled species or ecosystems, special licenses were not needed for field site access or research operations. The climate data (Fig 2) reveal that the area

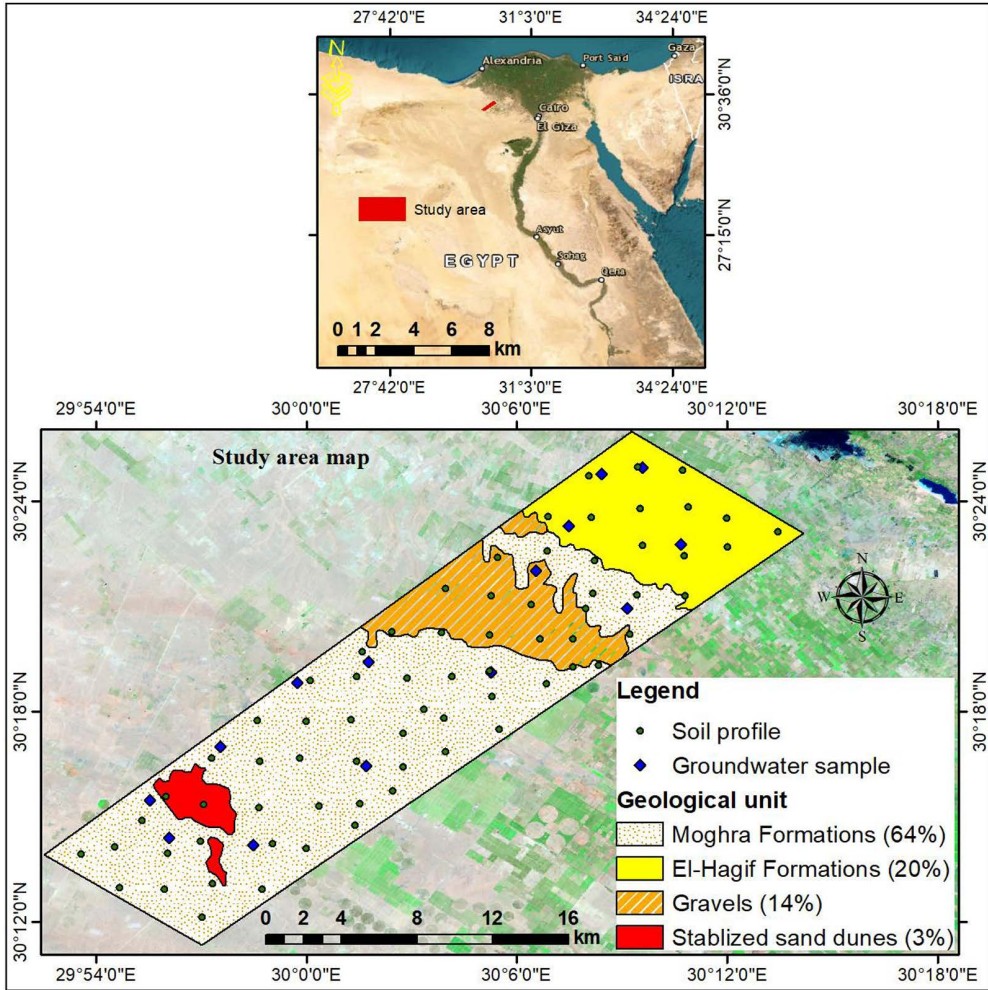

**Fig 1. Location maps of the studied area.**

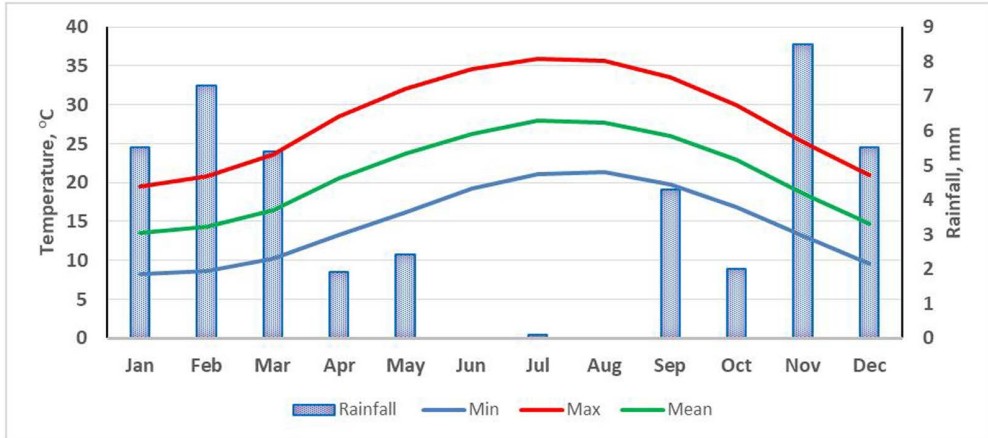

**Fig 2. Climate data of the studied region (average of ten years from 2010 to 2020).**

has Mediterranean climate conditions with arid summer and little rains in winter. January has a minimum temperature of 6°C, whereas August has a maximum of 33°C. The annual temperature averages 20°C and the total annual rainfall is 57 mm. Therefore, the soil has a "Thermic" temperature regime and a "Torric" moisture regime [50].

The digital elevation model (DEM of 12.5-m resolution) generated by the Advanced Land Observing Satellite-1 (ALOS) shows that the elevation varies between −14–212 m above sea level, while the slope gradient ranges from 0 to 96% (Fig 3). According to CONCO-Coral/EGPC [51], the geological formations vary from the Miocene to Quaternary eras (Fig 1). The lower Miocene Moghra Formations (sand, silt, and clay mixed with minor carbonate interbeds) cover 64% of the total area. The Late Tertiary (Pliocene) sediments represented by El Hagif Formations (limestone with marl inter-beds) occupy 20% of the total area. The Quaternary deposits cover 17% of the total area, including gravels (14%) and stabilized sand dunes (3%). The region is dominated by three land use/land cover patterns, including barren lands (69%), cultivated

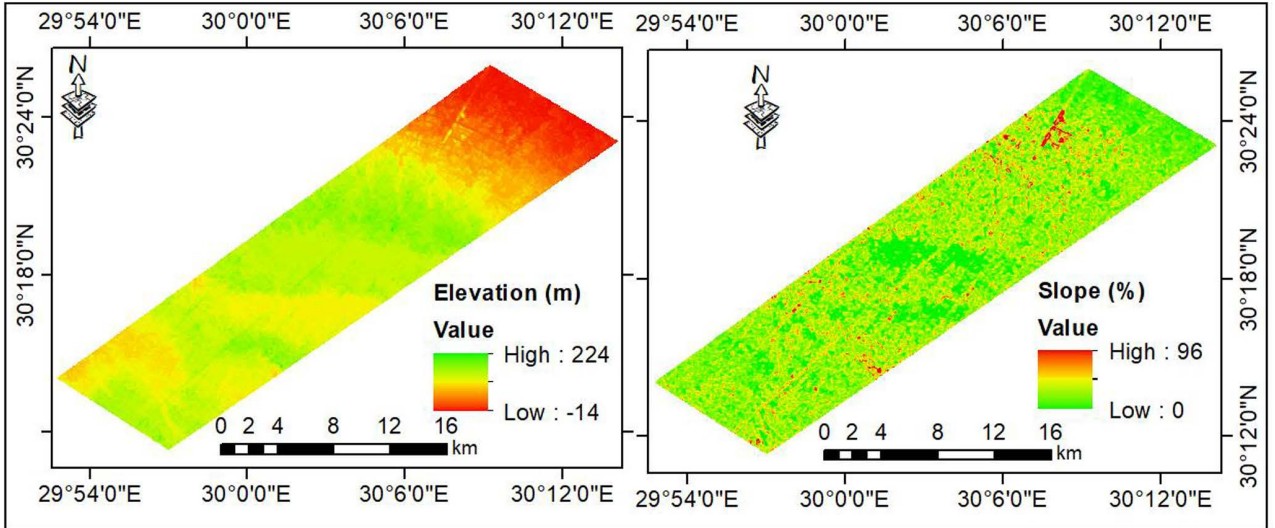

**Fig 3. Elevation and slope maps of the studied region.**

lands (40%), and urban areas (less than 1%). The soils are classified according to WRB system [52] as Arenosols (49%), Regosols (34%), Leptosols (11%), and Solonchaks (6%). The Moghra aquifer is the primary source of groundwater utilized for irrigation and other uses.

## 2.2. Field work and laboratory analysis

Seventy geo-referenced soil profiles (Fig 1) representing the geological units were dug to a depth of 150 cm or lithic contact. The morphological features of each profile were delineated according to Soil Science Division Staff [53]. Soil samples were collected from the subsequent horizons for further laboratory analysis. The soil analyses were executed following the standard methods outlined by Soil Survey Staff [54] and the main soil characteristics are presented in Table 1. In addition, fourteen groundwater samples were collected from artesian wells located nearby the sites of the collected soil samples. In situ measurements of pH and EC were conducted using a portable HACH instrument (HQ 40d, multi, USA). Thereafter, water samples were collected in 1 L high-density polypropylene bottles and transported to the laboratory for further analyses, following standard methods set by Rice, Baird [55]. The chemical composition of groundwater samples is given in Table 2.

## 2.3. Crop suitability modeling

To perform crop suitability analysis, four steps were executed as shown in Fig 4 as follows:

**2.3.1. Generation of thematic layers.** The sub-criteria determining climate, landscape, and groundwater suitability for the selected three crops were identified. The monthly climate data covering ten years from 2010 to 2020 were acquired from the WorldClim database (https://www.worldclim.org/). Thematic layers for minimum, maximum, and mean temperature during the growing periods for the selected crops were prepared. To obtain one value representing the whole profile, soil attributes for each horizon were recalculated considering the depth of the soil profile and weighting factors set by Sys, Van-Ranst [56]. Thereafter, raster layers for soil and groundwater characteristics were generated within ArcGIS 10.8 software using the inverse distance weighting (IDW) interpolation technique. The IDW is one of the simplest and most accurate interpolation techniques that is increasingly implemented in geosciences [57–60]. The IDW predicts the spatial value at the non-sampled site (Z(x)) using linear combinations of available data including the value of the

**Table 1. Descriptive statistics of main soil properties.**

| Property | Unit | Min | Max | Mean | SD | CV, % |
|---|---|---|---|---|---|---|
| Depth | cm | 10.00 | 180.00 | 127.07 | 43.85 | 34.51 |
| CF | % | 0.25 | 14.03 | 2.02 | 1.81 | 89.97 |
| pH | --- | 7.51 | 9.17 | 8.01 | 0.29 | 3.63 |
| EC | dS m$^{-1}$ | 0.24 | 54.17 | 6.06 | 8.17 | 134.83 |
| ESP | --- | 2.95 | 28.89 | 9.86 | 5.70 | 57.82 |
| CaCO$_3$ | g kg$^{-1}$ | 1.10 | 561.50 | 58.54 | 99.03 | 169.17 |
| Gypsum | | 2.45 | 36.74 | 20.26 | 8.56 | 42.24 |
| OC | | 0.63 | 4.72 | 2.03 | 0.92 | 45.56 |
| Sand | % | 41.68 | 96.03 | 78.40 | 12.24 | 15.62 |
| Silt | | 1.91 | 55.34 | 16.85 | 12.42 | 73.70 |
| Clay | | 0.00 | 13.90 | 4.75 | 2.99 | 62.86 |

CF, coarse fragments; EC; electrical conductivity; ESP, exchangeable sodium percentage; OC, organic carbon; SD, standard deviation; CV, coefficient of variation

**Table 2. Descriptive statistics of groundwater chemical composition.**

| Property | Unit | Min | Max | Mean | SD | CV, % |
|---|---|---|---|---|---|---|
| pH | --- | 7.12 | 8.85 | 7.98 | 0.39 | 4.83 |
| EC | dS m$^{-1}$ | 0.59 | 3.89 | 1.18 | 0.87 | 73.79 |
| Ca$^{2+}$ | mmol$_c$ L$^{-1}$ | 1.52 | 12.60 | 3.60 | 2.89 | 80.38 |
| Mg$^{2+}$ | | 0.42 | 5.05 | 1.81 | 1.33 | 73.43 |
| Na$^+$ | | 3.10 | 21.00 | 6.21 | 4.60 | 74.13 |
| K$^+$ | | 0.11 | 0.31 | 0.20 | 0.05 | 25.86 |
| Cl$^-$ | | 3.73 | 34.40 | 9.02 | 7.94 | 88.04 |
| SO$_4{}^{2-}$ | | 1.00 | 4.58 | 2.05 | 0.96 | 47.05 |
| HCO$_3{}^-$ | | 0.21 | 1.08 | 0.74 | 0.31 | 42.09 |
| SAR | --- | 2.51 | 7.06 | 3.66 | 1.10 | 30.10 |

SAR, sodium adsorption ratio

measured point (Zi), the total number of known points (n), the distance between predicted and measured sites (di), and parameter holding exponential value (β) as follows [61]:

$$Z_{(x)} = \frac{\sum_{i=1}^{n} Z_i \frac{i}{d_i^\beta}}{\sum_{i=1}^{n} \frac{1}{d_i^\beta}}$$

**2.3.2. Raster layer fuzzification.** The FMFs (fuzzy membership functions) integrated with ArcGIS 10.8 software were implemented to convert each pixel in the generated raster layers to a score between 0 and 1. As shown in Table 3, the applied functions were linear-positive (FLP) and linear-negative (FLN). The user-input maximum (U) and minimum (L) limits for a variable (x) are included to compute scores as follows [61]:

$$FLP = \begin{cases} 1 \; if \; x \geq U \\ \frac{x-L}{U-L} \; if \; L < x < U \\ 0 \; if \; x \leq L \end{cases}$$

$$FLN = \begin{cases} 1 \; if \; x \leq L \\ \frac{U-x}{U-L} \; if \; L < x < U \\ 0 \; if \; x \geq U \end{cases}$$

The lower and upper limits of environmental parameters related to crop requirements were identified according to global scientific resources [62–68]. The national standards suggested by MALR (Ministry of Agriculture and Land Reclamation) for producing wheat [69], maize [70], and broad bean [71] were also considered.

**2.3.3. Weight allocation.** The AHP procedure [30] was implemented to specify the weighting factors for main and sub-criteria related to site suitability. Firstly, a pairwise comparison model including three major criteria; climate, landscape, and groundwater was developed. Using a rating scale from 1 to 9, identifying the relative influence of each criterion to one another was done (Table 4). Additionally, pairwise comparison models (n × n) were also developed to compare the sub-criteria characterizing each main group. The judgments of ten local soil experts obtained through questionnaires besides the authors' experiences were utilized to set the relative importance of each criterion. The weight values for both the main and sub-criteria were calculated using the AHP online system package (AHP-OS). The questionnaire is given to

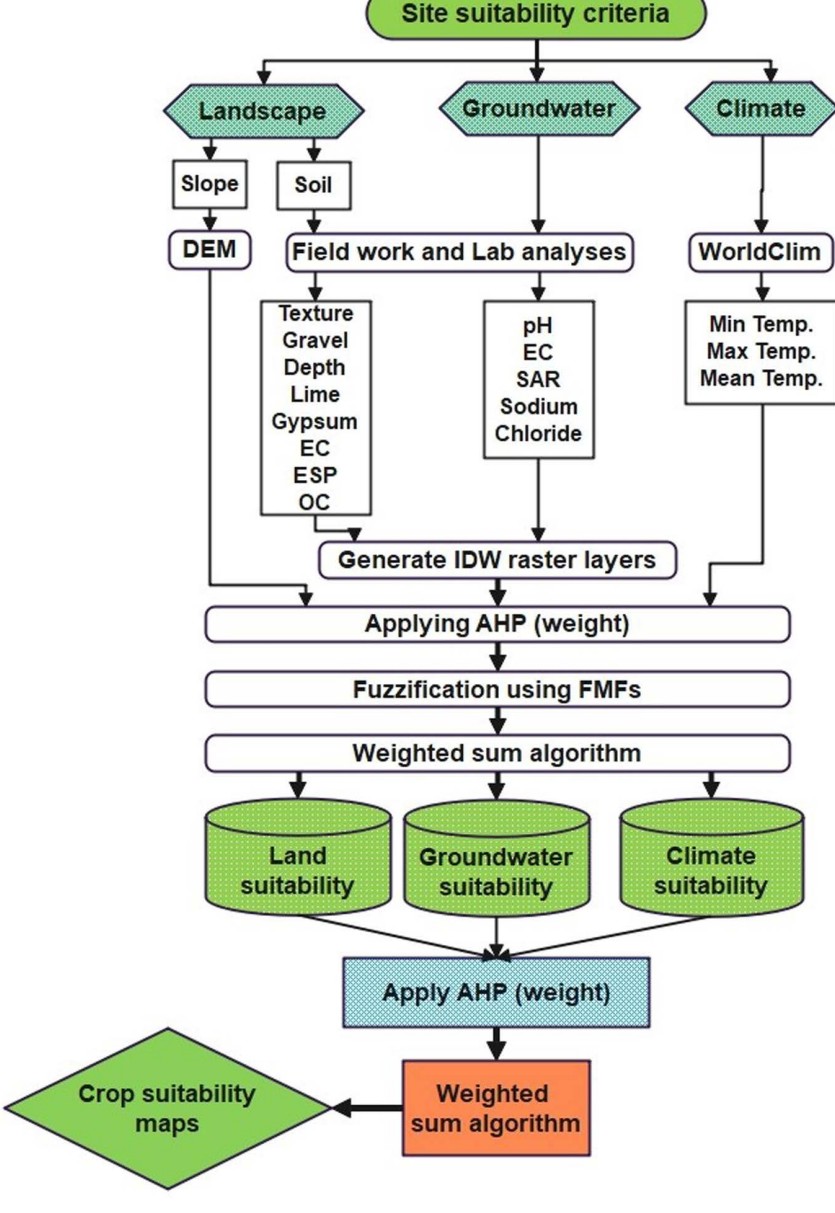

**Fig 4. Flowchart of the proposed methodology applied for modeling crop suitability.**

each expert's email address, the form is completed and returned to us. All specialists understand the nature of the current research, which does not include any human or animal experiments, thus it does not require specific approval. Consent was requested via email, with the agreement to participate explicitly mentioned in their written responses. This written consent, submitted and received by email, acts as documented informed consent.

To ensure the validity of the comparison models, the consistency ratio (CR) was checked as matrices achieving CR below 0.10 are acceptable. Moreover, to obtain high accuracy, the AHP was implemented twice using both the arithmetic as well as geometric mean algorithms of the experts' opinions. Consequently, the method demonstrated the lowest CR

**Table 3. Fuzzy membership functions (FMFs) applied for scoring crop suitability criteria.**

| Major criterion | Sub-criterion | Crop | FMF | Lower limit (L) | Upper limit (U) |
|---|---|---|---|---|---|
| Climate | Min temperature at growth period, °C | Wheat | Linear-positive | 6 | 10 |
| | | Broad bean | Linear-positive | 4 | 15 |
| | | Maize | Linear-negative | 18 | 30 |
| | Max temperature at growth period, °C | Wheat | Linear-negative | 20 | 32 |
| | | Broad bean | Linear-negative | 20 | 33 |
| | | Maize | Linear-negative | 24 | 40 |
| | Mean temperature at growth period, °C | Wheat | Linear-positive | 8 | 18 |
| | | Broad bean | Linear-positive | 10 | 25 |
| | | Maize | Linear-negative | 24 | 40 |
| Landscape | Slope, % | All | Linear-negative | 2 | 30 |
| | Sand, % | All | Linear-negative | 30 | 90 |
| | Clay, % | | Linear-positive | 0 | 25 |
| | Coarse fragments, % | All | Linear-negative | 5 | 80 |
| | Depth, cm | Wheat | Linear-positive | 10 | 90 |
| | | Broad bean | | 25 | 100 |
| | | Maize | | 20 | 100 |
| | CaCO$_3$, g kg$^{-1}$ | Wheat | Linear-negative | 30 | 600 |
| | | Broad bean | | 0 | 600 |
| | | Maize | | 0 | 350 |
| | Gypsum, g kg$^{-1}$ | All | Linear-negative | 0 | 200 |
| | pH | All | Linear-negative | 7 | 8.5 |
| | EC, dS m$^{-1}$ | Wheat | Linear-negative | 6 | 20 |
| | | Broad bean | | 1.5 | 12 |
| | | Maize | | 2 | 12 |
| | ESP | Wheat | Linear-negative | 15 | 45 |
| | | Broad bean | | 8 | 25 |
| | | Maize | | 8 | 25 |
| | OC, g kg$^{-1}$ | Wheat | Linear-positive | 0 | 6 |
| | | Broad bean | | 0 | 8 |
| | | Maize | | 0 | 8 |
| Irrigation water | pH | All | Linear-negative | 6.5 | 8.4 |
| | EC, dS m$^{-1}$ | Wheat | Linear-negative | 4 | 8.7 |
| | | Broad bean | | 1.1 | 4.5 |
| | | Maize | | 1.1 | 3.9 |
| | SAR | All | Linear-negative | 3 | 20 |
| | Na$^+$, and Cl$^-$, mmol$_C$ L$^{-1}$ | Wheat | Linear-negative | 10 | 20 |
| | | Broad bean | | 0 | 3 |
| | | Maize | | 10 | 20 |

was considered for further analysis. The CR is computed based on CI (consistency index) and RI (random consistency index), $\lambda_{max}$ (highest eigenvalue for each comparison model), and n (number of criteria) using the following equations [30]:

$$CR = \frac{CI}{RI}$$

**Table 4. The evaluation scale used in the analytical hierarchical process.**

| Importance degree | Definition |
|---|---|
| 1 | Equal importance |
| 3 | Moderate importance |
| 5 | Essential or strong importance |
| 7 | Demonstrated importance |
| 9 | Absolute importance |
| 2–4–6–8 | Intermediate values |

$$CI = \frac{(\lambda_{max} - 1)}{(n - 1)}$$

The RI values are identified based on the number of analyzed criteria as illustrated in Table 5.

**2.3.4. Developing crop suitability maps.** The weighted sum algorithm was employed to integrate the fuzzified raster layers with their respective weight, which were previously determined from the AHP under the GIS environment. Five suitability classes were identified: highly suitable (S1 ⟩ 0.80), moderately suitable (S2 0.8–0.6), marginally suitable (S3 0.6–0.4), and currently not suitable (N1 0.4–0.2), and permanently not-suitable (N2 ⟨ 0.20).

## 3. Results

### 3.1. Climate suitability

The results of AHP applied to assess interactions among main and sub-criteria associated with site suitability analyses are presented in Table 6. The atmosphere temperature (minimum, mean, and maximum) was considered to evaluate the fitness of climate conditions for the selected crops. Accordingly, for winter crops such as wheat and broad bean, the minimum temperature received the first rank. It accounted for 63% of climate suitability, followed by mean temperature at 24%, while maximum temperature contributed to the least, i.e., 14%. In contrast, for summer crops such as maize, the maximum temperature emerged as the most substantial determinant and contributed to 62% of climate suitability, followed by mean temperature (27%), while the minimum temperature contributed to the least, i.e., 12%. The climate suitability results, illustrated in Fig 5, indicate that the suitability index ranged from 0.76 to 0.92 for wheat and from 0.72 to 0.82 for broad bean. Thus, the studied region was classified as S1 and S2 for both wheat and broad bean. Compared with wheat, broad bean is highly preferable as these classes encompass 85 and 15% of the total area, respectively (Table 7). For wheat, the S1 and S2 classes represented 60 and 40% of the total area, respectively. The optimum climate conditions for winter crops were mainly found in the southwestern parts of the studied region and declined towards the northeast. Conversely, the climate suitability index for maize ranged between 0.61 to 0.70, indicating that the studied region would be marginally suitable for maize production.

### 3.2. Landscape suitability

As shown in Table 6, among eleven criteria affecting landscape (terrain and soil) suitability for crop production, soil salinity expressed as EC was the most important determinant with a specific contribution of 23%, followed by sodicity expressed

**Table 5. The number of criteria (n) and random consistency index (RI) values.**

| n | 1 | 2 | 3 | 4 | 5 | 6 | 7 | 8 | 9 | 10 | 11 | 12 | 13 |
|---|---|---|---|---|---|---|---|---|---|---|---|---|---|
| RI | 0.00 | 0.00 | 0.58 | 0.90 | 1.12 | 1.24 | 1.32 | 1.41 | 1.45 | 1.51 | 1.52 | 1.54 | 1.56 |

**Table 6. Pairwise comparison models and weights of criteria adopted in crop suitability analysis.**

| Criterion | Pairwise comparison matrix | | | | | | | | | | | Rank | weight |
|---|---|---|---|---|---|---|---|---|---|---|---|---|---|
| | (1) | (2) | (3) | (4) | (5) | (6) | (7) | (8) | (9) | (10) | (11) | | |
| Climate (1) | 1 | 1/3 | 1/4 | | | | | | | | | 3 | 0.126 |
| Land (2) | 3 | 1 | 1 | | | | | | | | | 2 | 0.416 |
| Irrigation water (3) | 4 | 1 | 1 | | | | | | | | | 1 | 0.458 |
| $\lambda_{max}=3.009$; n = 3; RI = 0.58; CI = 0.045; CR (CI/ RI) = 0.007 | | | | | | | | | | | | Sum | 1.000 |
| Climate suitability for winter crops | | | | | | | | | | | | | |
| (1) Min temperature | 1 | 4 | 3 | | | | | | | | | 1 | 0.625 |
| (2) Max temperature | 1/4 | 1 | 1/2 | | | | | | | | | 3 | 0.137 |
| (3) Mean temperature | 1/3 | 2 | 1 | | | | | | | | | 2 | 0.238 |
| $\lambda_{max}=3.018$; n = 3; RI = 0.58; CI = 0.009; CR (CI/ RI) = 0.015 | | | | | | | | | | | | Sum | 1.000 |
| Climate suitability for summer crops | | | | | | | | | | | | | |
| (1) Min temperature | 1 | 1/4 | 1/3 | | | | | | | | | 3 | 0.117 |
| (2) Max temperature | 4 | 1 | 3 | | | | | | | | | 1 | 0.615 |
| (3) Mean temperature | 3 | 1/3 | 1 | | | | | | | | | 2 | 0.268 |
| $\lambda_{max}=3.074$; n = 3; RI = 0.58; CI = 0.037; CR (CI/ RI) = 0.064 | | | | | | | | | | | | Sum | 1.000 |
| Land suitability | | | | | | | | | | | | | |
| (1) Slope | 1 | 1/2 | 1/2 | 3 | 3 | 2 | 1/4 | 1/4 | 2 | 1/3 | 1 | 6 | 0.064 |
| (2) Coarse fragments | 2 | 1 | 1/3 | 2 | 2 | 2 | 1/4 | 1/4 | 2 | 2 | 4 | 4 | 0.087 |
| (3) Depth | 2 | 3 | 1 | 2 | 2 | 3 | 1/4 | 1/4 | 2 | 2 | 3 | 3 | 0.108 |
| (4) Sand | 1/3 | 1/2 | 1/2 | 1 | 1/2 | 1/2 | 1/4 | 1/4 | 1/2 | 1/3 | 1/2 | 11 | 0.032 |
| (5) Clay | 1/3 | 1/2 | 1/2 | 2 | 1 | 2 | 1/4 | 1/4 | 1/2 | 1/3 | 2 | 8 | 0.047 |
| (6) pH | 1/2 | 1/2 | 1/3 | 2 | 1/2 | 1 | 1/4 | 1/4 | 1/2 | 1 | 1 | 9 | 0.042 |
| (7) EC | 4 | 4 | 4 | 4 | 4 | 4 | 1 | 2 | 4 | 3 | 4 | 1 | 0.234 |
| (8) ESP | 4 | 4 | 4 | 4 | 4 | 4 | 1/2 | 1 | 4 | 3 | 4 | 2 | 0.207 |
| (9) OC | 1/2 | 1/2 | 1/2 | 2 | 2 | 2 | 1/4 | 1/4 | 1 | 1/2 | 2 | 7 | 0.055 |
| (10) CaCO$_3$ | 3 | 1/2 | 1/2 | 3 | 3 | 1 | 1/3 | 1/3 | 2 | 1 | 3 | 5 | 0.086 |
| (11) Gypsum | 1 | 1/4 | 1/3 | 2 | 1/2 | 1 | 1/4 | 1/4 | 1/2 | 1/3 | 1 | 10 | 0.038 |
| $\lambda_{max}=11.938$; n = 11; RI = 1.52; CI = 0.094; CR (CI/ RI) = 0.062 | | | | | | | | | | | | Sum | 1.000 |
| Irrigation water suitability | | | | | | | | | | | | | |
| (1) pH | 1 | 1/4 | 1/3 | 1/4 | 1/4 | | | | | | | 5 | 0.058 |
| (2) EC | 4 | 1 | 2 | 2 | 2 | | | | | | | 1 | 0.338 |
| (3) SAR | 3 | 1/2 | 1 | 1/3 | 1/3 | | | | | | | 4 | 0.118 |
| (4) Na | 4 | 1/2 | 3 | 1 | 2 | | | | | | | 2 | 0.276 |
| (5) Cl | 4 | 1/2 | 3 | 1/2 | 1 | | | | | | | 3 | 0.210 |
| $\lambda_{max}=5.256$; n = 5; RI = 1.21; CI = 0.066; CR (CI/ RI) = 0.055 | | | | | | | | | | | | Sum | 1.000 |

as ESP at 21% and rooting area expressed as soil depth at 11%, while sand content contributed to only 3%. The land suitability analysis, presented in Fig 6, reveals that the land suitability index for wheat ranged from 0.46 to 0.90, indicating that the studied region was classified as suitable (S1, S2, and S3). As shown in Table 7, areas characterized as S1, S2, and S3 covered 74, 24, and 2% of the total area, respectively. The optimum land suitability conditions were mainly found in the inner parts and decreased towards the outer parts. On the other hand, the land suitability index for both broad bean and maize varied between 0.31 and 0.88, indicating a wide suitability range from S1 to N1. For broad bean, areas classified as S1, S2, S3, and N1 accounted for 26, 55, 19, and 1% of the total area, respectively. For maize, 29, 51, 19, and 1% of the total area occurred in these classes, respectively. The favorable suitability conditions for broad bean and maize were

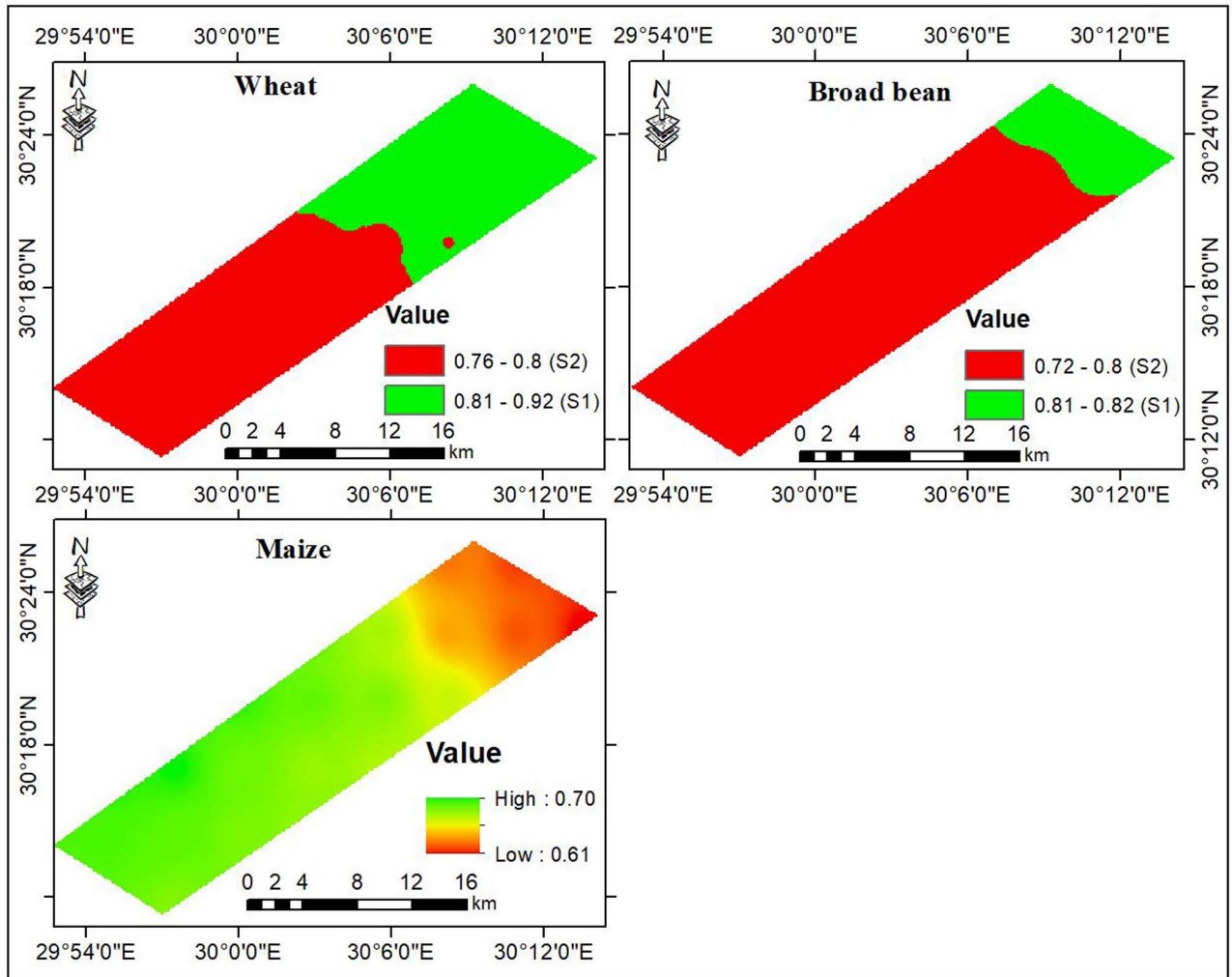

**Fig 5. Climate suitability for the selected crops.**

mainly observed in the middle parts; meanwhile, the worst levels were mainly found in the northeastern and southwestern parts of the studied region besides small scattered patches in the middle parts.

### 3.3. Groundwater suitability

Results in Table 6 reveal that water quality parameters displayed various contributions to groundwater suitability for sprinkler irrigation. Among these parameters, potential salinity hazards quantified by EC ranked as the first most important parameter with a specific effect of 34% followed by toxicity hazards linked to sodium and chloride ions with specific contributions of 28 and 21%, respectively. On the other hand, pH received the least significant contribution of only 6%. Results in Fig 7 demonstrate that the water suitability index ranged from 0.47 to 0.96 for wheat, implying that groundwater was suitable (S1, S2, and S3) for irrigating wheat under center pivot systems. As presented in Table 7, the suitability classes were distributed as 93% S1, 5% S2, and 2% S3. The suitability index for maize varied between 0.13 to 0.96, indicating a wide suitability range from S1 to N2. However, groundwaters in 88% of the total area fitted the S1 class, while

**Table 7. Areas of suitability classes in the studied region.**

| Suitability factor | Crop | Area | Suitability class | | | | |
|---|---|---|---|---|---|---|---|
| | | | S1 | S2 | S3 | N1 | N2 |
| Climate | Wheat | ha | 18145.00 | 12084.00 | --- | --- | --- |
| | | % | 60.03 | 39.97 | --- | --- | --- |
| | Borad bean | ha | 25740.00 | 4489.00 | --- | --- | --- |
| | | % | 85.15 | 14.85 | --- | --- | --- |
| | Maize | ha | --- | --- | 30229.00 | --- | --- |
| | | % | --- | --- | 100.00 | --- | --- |
| Landscape | Wheat | ha | 22514.00 | 7201.00 | 514.00 | --- | --- |
| | | % | 74.48 | 23.82 | 1.70 | --- | --- |
| | Borad bean | ha | 7732.00 | 16594.00 | 5710.00 | 193.00 | --- |
| | | % | 25.58 | 54.89 | 18.89 | 0.64 | --- |
| | Maize | ha | 8859.00 | 15406.00 | 5689.00 | 275.00 | |
| | | % | 29.31 | 50.96 | 18.82 | 0.91 | --- |
| Irrigation water | Wheat | ha | 28058.00 | 1539.00 | 632.00 | --- | --- |
| | | % | 92.82 | 5.09 | 2.09 | --- | --- |
| | Borad bean | ha | --- | --- | 26239.00 | 3941.00 | 49.00 |
| | | % | --- | --- | 86.80 | 13.04 | 0.16 |
| | Maize | ha | 26498.00 | 2195.00 | 881.00 | 496.00 | 159.00 |
| | | % | 87.66 | 7.26 | 2.91 | 1.64 | 0.53 |
| Overall suitability | Wheat | ha | 27305.00 | 2924.00 | --- | --- | --- |
| | | % | 90.33 | 9.67 | --- | --- | --- |
| | Borad bean | ha | --- | 16122.00 | 14107.00 | --- | --- |
| | | % | --- | 53.33 | 46.67 | --- | --- |
| | Maize | ha | 16602.00 | 12839.00 | 788.00 | --- | --- |
| | | % | 54.92 | 42.47 | 2.61 | --- | --- |

the remaining area encompassed the lower suitability categories; S2 (7%), S3 (3%), N1 (2%), and N2 (1%). The worst suitability conditions for both wheat and maize were mainly found in small patches in the northeastern parts of the studied region. The suitability index for broad bean varied between 0.19 to 0.48, indicating that groundwater in the studied region was classified as marginally suitable (S3) and not suitable (N1 and N2). Results in Table 7 illustrate that groundwater classified as S3, N1, and N2 accounted for 87, 13, and less than 1% of the total area, respectively.

### 3.4. Overall site suitability

The AHP results, given in Table 6, illustrate that irrigation water quality emerged as the most substantial factor affecting crop production in the studied region, contributing to 46%, followed by landscape characteristics at 42% and climate conditions at 12%. The overall crop suitability maps, shown in Fig 8, demonstrate that the site suitability index for wheat ranged from 0.66 to 0.91, indicating that the studied region was classified as S1 and S2. As shown in Table 7, the S1 class dominated the region, accounting for 90% of the total area. The S2 class occupied only 10% of the total area and occurred in small patches across the studied region. The suitability index for broad bean varied from 0.44 to 0.67, implying that the studied region was classified as S2 and S3. The S2 class represented 53% of the total area and dominated the middle and northeastern parts of the studied region. The S3 class accounted for 47% of the total area and occurred in the northeastern and southwestern parts of the studied region. For maize, the suitability index ranged between 0.47 to 0.88, indicating that the studied region was classified as suitable (S1, S2, and S3). The S1 class accounted for 55% of the total

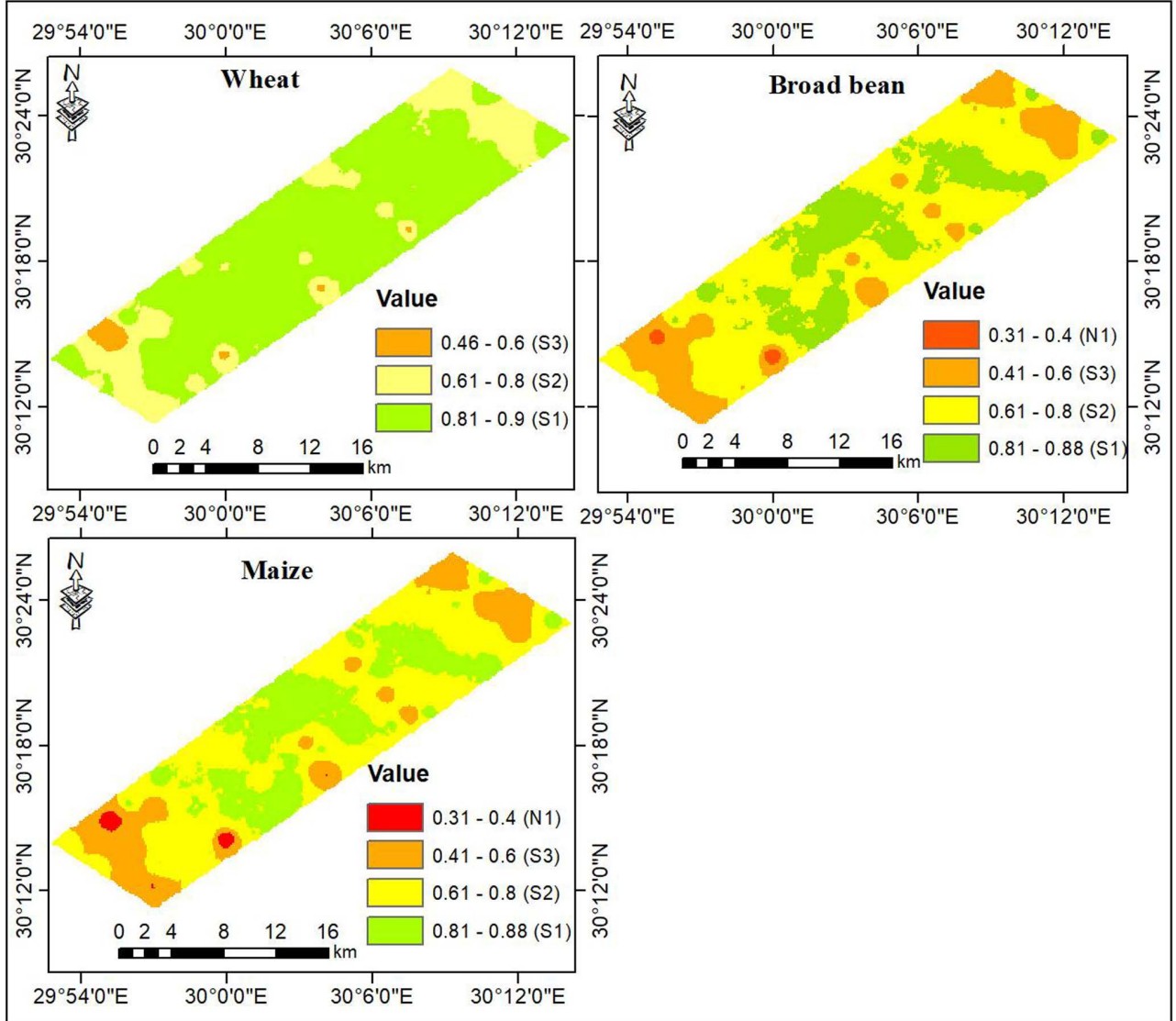

**Fig 6. Land suitability for the selected crops.**

area and was mainly found in the middle parts. The S2 class encompassed 43% of the total area and occurred primarily in the northeastern and southwestern parts of the studied region. The S3 class represented only 3% of the total area and occurred in a small patch in the northeastern part of the studied region. Based on the crop suitability maps, among the selected three crops, wheat is the most preferable field crop in the studied region followed by maize, while broad bean ranked as the third.

## 4. Discussion

In irrigated croplands, especially in arid and semiarid regions, the appropriateness for climate conditions of crop cultivation is closely related to atmosphere temperatures (minimum, mean, and maximum) during the growing periods [67,68]. The temperature directly affects the physicochemical processes within plants and control their growth rates and biomass

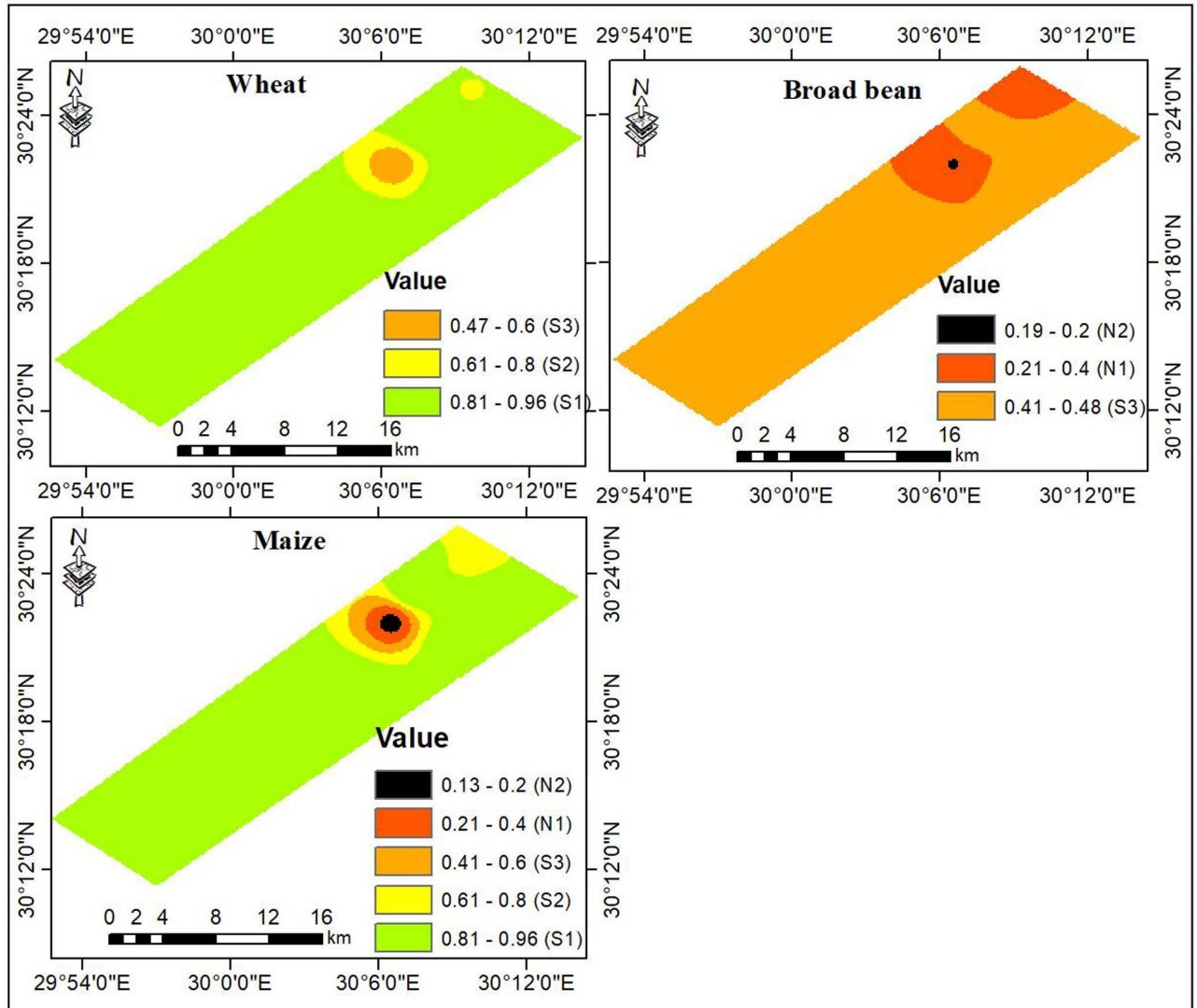

**Fig 7. Groundwater suitability for the selected crops.**

production [72]. Among several climate factors, atmosphere temperature is expected to receive too more attentions owing to the ongoing global warming and accelerated climate change [73]. In addition, other factors governing crop evapotranspiration (ET) like solar radiation, relative humidity, wind speed, and sunshine hours can be easily managed using effective irrigation scheduling and proper agronomic practices [74]. In the studied region, atmosphere temperature would meet growth requirements for winter crops such as broad bean and wheat. However, high temperature during the summer season renders debates concerning maize cultivation. Heat stress negatively affects maize productivity since it inhibits seed germination, impairs seedling growth, retards flower differentiation, and ultimately reduces grain yield and quality [75,76].

The land resources, including terrain soil profile characteristics quantify landscape suitability for crop cultivation [77]. Based on the AHP provided in our work, EC, ESP, and soil depth had the gre`atest effects on land suitability for crop production. Generally, in AHP-based crop suitability studies, the weight of each prosperity is directly linked to the predominant local conditions expressed by expert judgments. For instance, suitability models for cultivating maize [67] and wheat [68]

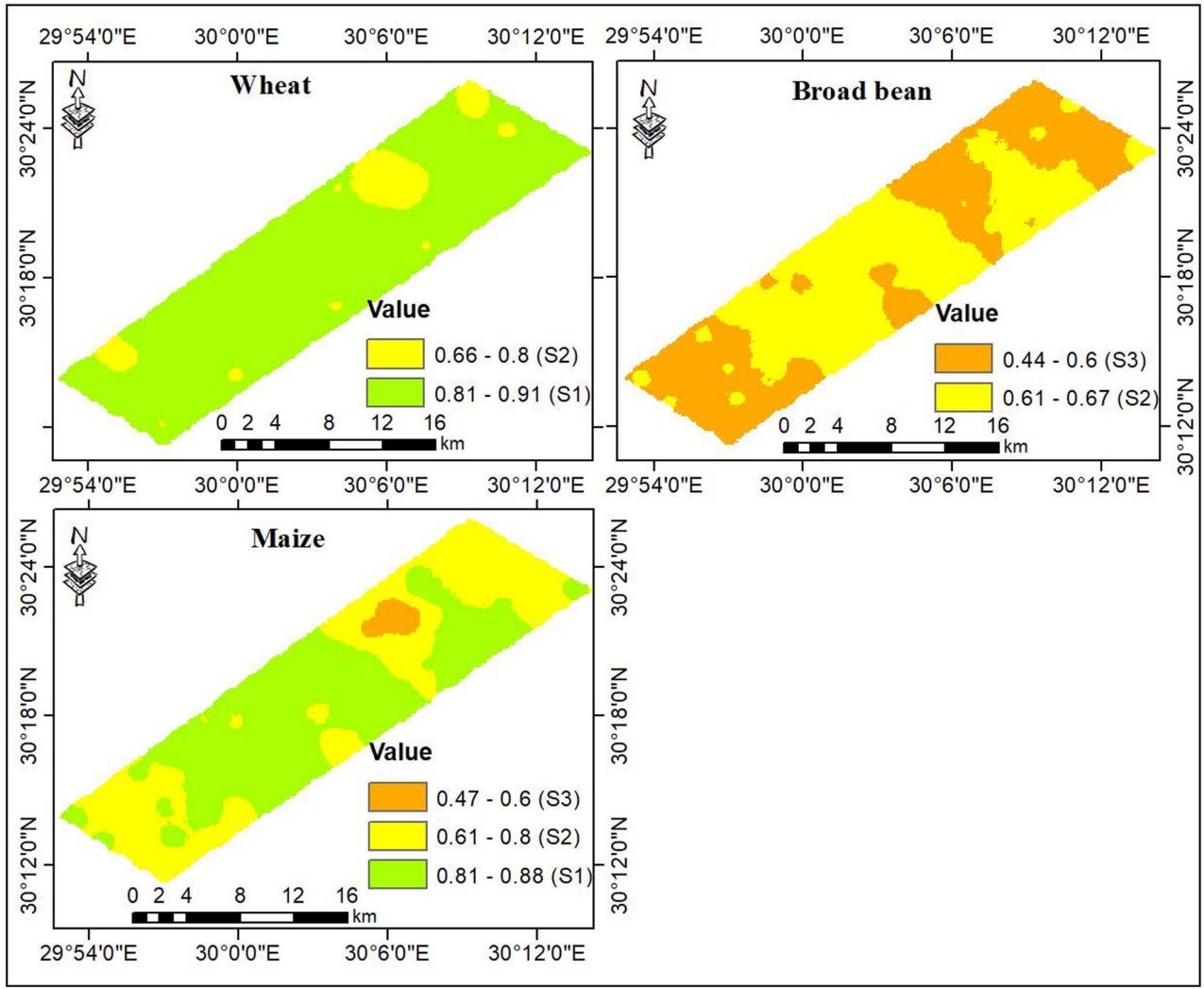

**Fig 8. Overall suitability maps for the selected crops.**

in a semi-arid region of Iran suggested that soil texture, pH, EC, and ESP were the most determinants for land suitability. Moreover, Günal, Kılıç [78] reported that soil texture and pH were the most important parameters affecting wheat production in a semi-arid region of Turkey. In Egypt, AHP applied to predict wheat suitability in El-Minia Governorate [79] revealed that soil depth exerted the highest impact. Moreover, suitability models developed using AHP weights [15] proved that EC, ESP, and CaCO$_3$ were the most influential parameters affecting land suitability for field crop production in the west Nile Delta region. Regarding major crop suitability criteria, our research highlighted he highest priority of groundwater quality over landscape and climate factors, which has been documented for an adjacent area [15]. Conversely, soil properties were more influential than irrigation water salinity and slope factors in assessing site suitability for wheat in the Upper Egypt [79].

The different land suitability status for the selected crops in the studied region is attributed mainly to large variations in terrain and key soil properties such as EC and ESP. In addition, the extent of land suitability is linked also to cropping type. For instance, wheat is a salt-tolerant crop, which can survive at high levels of salt and exchangeable sodium [80].

Therefore, around three-quarters of the studied region fitted the S1 class for wheat production. On the other hand, both maize and broad bean are classified as salt-sensitive crops, and thus the extent of land suitability was closely associated with salinity distribution across the studied region.

Groundwater irrigation is a fundamental component of sustainable food production in the drylands [9,81]. Yet, the efficacy of this practice is closely related to the sort and concentration of dissolved constituents, which affect soil-plant relationships under a specific irrigation system [82]. Quantifying these substances is essential to evaluate potential risks associated with irrigation water quality like salinity, infiltration, and toxicity [80]. Thus, in the current work, parameters determining these problems were considered to assess the potential of groundwater irrigation using the center pivot apparatus. Accordingly, the AHP assigned the highest weights for EC, Na +, and Cl⁻. These findings are in line with those reported by Abuzaid and El-Husseiny (15) in the west Nile Delta region. The groundwater resources in the studied area are preferable for irrigating wheat and maize using center pivot systems. Although maize is a salt-sensitive crop, it can tolerate high concentrations of $Na^+$ and $Cl^-$ in saline sprinkling waters [80].

The overall crop suitability maps highlighted that the studied region exhibited potential for cultivating the three crops. These findings are rather similar to those derived from applying the Applied Land Evaluation System for arid and semi-arid regions (ALESarid-GIS model) in the western fringes of the Nile Delta. For instance, the study conducted in Wadi El-Natrun District [83] revealed that the region fell in the S2, S3, and S4 (conditionally suitable) for wheat, while in the S2, S3, S4, and N1 classes for maize and broad bean. Moreover, certain areas of the New Delta region were classified as S1, S2, and S4 for wheat production, while as S2, S3, and S4 for maize cultivation [84]. Based on a hybrid AHP-GIS approach [15], a newly developed area in El-Saddat City was classified as S2 and S3 for the three crops. The ALES-GIS model supposes equal weights and neglects relative importance of different factors affecting crop performance. Although the AHP-GIS framework has appointed the specific weight of each environmental covariate, the supervised rating of suitability classes is a major drawback. Hence, the multi-criteria analysis provided in our research could address this issue to introduce a deep analysis of crop requirements. Yet, optimal site suitability evaluations impose a more detailed representation of soil and groundwater resources through intensifying the sampling density. Therefore, for future research, it is essential to digital maps with appropriate spatial resolutions.

Generally, the varied suitability patterns for the selected crops in the investigated region is directly linked to both type and intensity of constraints affecting plant growth and crop yield [19,20]. Normally, zones having no or slight constraints exert higher suitability levels compared with areas suffering moderate, severe, or very severe limitations [32]. The studied region is highly potential for wheat cultivation since 90% of the total area was classified as S1. This, in turn, reflects favorable climate conditions, landscape properties, and irrigation water quality that met optimal wheat requirements. For broad bean cultivation, the studied area undergoes moderate to severe groundwater limitations and slight to moderate landscape limitations. Consequently, the suitability levels declined to be in the S2 and S3 classes. For maize production, the greatest portion of the studied region (88%) has no restrictions related to groundwater irrigation, the landscape characteristics exhibited moderate to slight limitations, and the climate conditions would pose moderate constraints. Accordingly, the suitability levels were arranged in the S1, S2, and S3 classes.

## 5. Conclusion

In the current work, we provided a new framework based on integrating GIS-fuzzy sets with AHP to deeply analyze site suitability for cultivating wheat, broad bean, and maize under center pivot irrigation systems. The trial was conducted in a newly reclaimed desert area in the western Nile Delta fringes, Egypt. The study revolved on matching local climate, landscape, and groundwater qualities with crop requirements. The climate conditions were highly suitable and suitable for cultivating wheat and broad bean but marginally suitable for maize production. The AHP-based weights assigned the highest priorities for soil salinity, sodicity, and effective depth in landscape suitability for crop production. Moreover, parameters such as EC, Na, and Cl⁻ had substantial effects on groundwater potentiality for sprinkler irrigation. The land

resources in the studied region are suitable (S1, S2, and S3) for wheat cultivation; meanwhile, the suitability classes fitted the S1 to N1 classes for broad bean and maize production. Groundwater in the studied area would be preferable for irrigating wheat and maize under center pivot systems, while broad bean irrigation would pose potential toxicity problems. Among the three crops, wheat is the most recommended crop, where 90 and 10% of the total area fell in the S1 and S2 classes, respectively. Maize emerged as the second suitable crop with S1, S2, and S3 classes accounting for 55, 42, and 3% of the total area, respectively. Broad bean ranked third, where 53 and 47% of the total area were allocated in the S2 and S3 classes, respectively. The integrated use of AHP and GIS-fuzzy logic models would increase insight into linking locally dominant conditions with plant growth and crop yield. Our results would serve as a starting point for integrated land and groundwater management practices and for suitable food crop production in arid croplands. Yet, further investigations through increasing the density of groundwater samples are fundamentals for a deep depiction and better representation for the local limitations. The study advocates future crop-based suitability evaluations in order to verify the robustness of the developed methodology across various geographic locations and climate conditions.

## Acknowledgments

The authors extend their sincere appreciation to the RUDN University.

## Author contributions

**Conceptualization:** Ahmed S. Abuzaid, Hassan H. Abbas, Mostafa A. Mostafa, Yousif K. El Ghonamy, Nazih Y. Rebouh, Mohamed S. Shokr.

**Data curation:** Ahmed S. Abuzaid, Hassan H. Abbas, Mostafa A. Mostafa, Yousif K. El Ghonamy, Mohamed S. Shokr.

**Formal analysis:** Ahmed S. Abuzaid, Hassan H. Abbas, Mostafa A. Mostafa, Yousif K. El Ghonamy, Nazih Y. Rebouh, Mohamed S. Shokr.

**Investigation:** Ahmed S. Abuzaid, Mostafa A. Mostafa, Mohamed S. Shokr.

**Methodology:** Ahmed S. Abuzaid, Mostafa A. Mostafa, Yousif K. El Ghonamy, Mohamed S. Shokr.

**Project administration:** Mohamed S. Shokr.

**Resources:** Ahmed S. Abuzaid, Mostafa A. Mostafa.

**Software:** Ahmed S. Abuzaid, Mostafa A. Mostafa.

**Supervision:** Hassan H. Abbas, Nazih Y. Rebouh.

**Validation:** Mostafa A. Mostafa.

**Visualization:** Yousif K. El Ghonamy, Mohamed S. Shokr.

**Writing – original draft:** Ahmed S. Abuzaid, Hassan H. Abbas, Mostafa A. Mostafa, Yousif K. El Ghonamy, Nazih Y. Rebouh, Mohamed S. Shokr.

**Writing – review & editing:** Ahmed S. Abuzaid, Mohamed S. Shokr.

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
