## [Decision Letter · Decision Letter 0]

PONE-D-25-16781A comprehensive crop suitability assessment under modern irrigation system in arid croplandsPLOS ONE

Dear Dr. Shokr,

Thank you for submitting your manuscript to PLOS ONE. After careful consideration, we feel that it has merit but does not fully meet PLOS ONE’s publication criteria as it currently stands. Therefore, we invite you to submit a revised version of the manuscript that addresses the points raised during the review process.

Thank you for your submission to our journal. After careful review, the referees have provided valuable feedback that requires revision of your manuscript.

I kindly ask you to revise your article in accordance with the reviewers’ comments and suggestions. Once the revisions are complete, please resubmit your manuscript for further consideration.

If you have any questions, feel free to contact us.

We look forward to receiving your revised manuscript.

Kind regards,

Somayeh Soltani-Gerdefaramarzi, Ph. D.

Academic Editor

PLOS ONE

Journal Requirements:

2. You indicated that ethical approval was not necessary for your study. We understand that the framework for ethical oversight requirements for studies of this type may differ depending on the setting and we would appreciate some further clarification regarding your research. Could you please provide further details on why your study is exempt from the need for approval and confirmation from your institutional review board or research ethics committee (e.g., in the form of a letter or email correspondence) that ethics review was not necessary for this study? Please include a copy of the correspondence as an "Other" file.

4. In your Methods section, please provide additional information regarding the permits you obtained for the work. Please ensure you have included the full name of the authority that approved the field site access and, if no permits were required, a brief statement explaining why.

5. Thank you for stating the following in your Competing Interests section: “No”

6. We note that your Data Availability Statement is currently as follows: All relevant data are within the manuscript and its Supporting Information files

**Additional Editor Comments:**

Thank you for your submission to our journal. After careful review, the referees have provided valuable feedback that requires revision of your manuscript.

I kindly ask you to revise your article in accordance with the reviewers’ comments and suggestions. Once the revisions are complete, please resubmit your manuscript for further consideration.

If you have any questions, feel free to contact us.

Reviewers' comments:

Reviewer's Responses to Questions

**Comments to the Author**

1. Is the manuscript technically sound, and do the data support the conclusions?

Reviewer #1: Yes

Reviewer #2: Yes

2. Has the statistical analysis been performed appropriately and rigorously? 

Reviewer #1: Yes

Reviewer #2: Yes

3. Have the authors made all data underlying the findings in their manuscript fully available?

Reviewer #1: Yes

Reviewer #2: Yes

4. Is the manuscript presented in an intelligible fashion and written in standard English?

Reviewer #1: Yes

Reviewer #2: Yes

5. Review Comments to the Author

Reviewer #1: This article has made significant contributions in the field of crop suitability assessment in arid regions. The methods are scientific and the results have practical significance. However, it needs further improvement in data presentation, detailed description of the methods and writing norms. Overall, the quality is above average and it is suitable to be used as a reference basis for regional agricultural planning. The detailed opinions are as follows.

1. Keywords can include terms such as arid croplands.

2. In the introduction, the background of climate change and water resource shortage is mentioned, but the uniqueness of the study area and its impact on agriculture can be more specifically described. Data or references on the specific water resource situation and agricultural development challenges in the western Nile Delta of Egypt can be added.

3. In the methods section, when describing fuzzy logic, a more detailed explanation of how the upper and lower limit values are selected and how they are related to crop demands can be provided.

4. The data presentation in the results section needs to be more intuitive. Tables or charts can be used to enhance readability. For example, the distribution ratio of different crop suitability grades can be transformed into bar charts or pie charts.

5. The discussion section needs to make a more in-depth comparison of the results with existing studies, especially with the research results of other arid regions. In addition, the limitations of the method, such as sample size and spatial resolution, can be discussed, as well as the directions for future research.

6. The use of professional terms is inconsistent, with "center pivot" and "central pivot" being used interchangeably.

Reviewer #2: 1. What was the selection of criteria or factors based on? Please explain.

2. In the discussion section, it is better to make comparisons with the researches of others and the research done, and for the presented arguments, be sure to use valid and up-to-date references.

3. The advantages and disadvantages of the research done should be said.

4. Please provide appropriate and valid references for all provided relationships.

5. In the AHP method, did you use the opinions of relevant experts in the form of a questionnaire to determine the importance of the factors in the pairwise comparison matrix?

6. Please use the papers (https://doi.org/10.1007/s10661-022-10327-x,
https://doi.org/10.1080/03650340.2018.1549363,
https://doi.org/10.1016/j.geoderma.2017.09.012,
https://doi.org/10.1016/j.geoderma.2019.05.046,
https://doi.org/10.1080/00103624.2022.2072511;
https://doi.org/10.1080/03067319.2020.1746775;
https://doi.org/10.1007/s10661-022-10659-8;
https://doi.org/10.1080/00103624.2019.1626870) to improve the quality of the manuscript and use and add them to improve the quality of the manuscript, especially the introduction and discussion of the manuscript, description and interpretation of properties and select the criteria.

7. Please give the names of soils according to the WRB system.

8. What was the accuracy of the methods used? By which criteria are the methods evaluated?

9. Please check the grammar of the whole text with a native speaker and fix the errors.

6. PLOS authors have the option to publish the peer review history of their article (what does this mean? ). If published, this will include your full peer review and any attached files.

**Do you want your identity to be public for this peer review?** For information about this choice, including consent withdrawal, please see our Privacy Policy .

Reviewer #1: No

Reviewer #2: No

---

## [Author Response · Author response to Decision Letter 1]

21 May 2025

Response to Reviewer 1

Comment 1. Keywords can include terms such as arid croplands.

Response: The arid cropland has been added to the keywords

Comment 2. In the introduction, the background of climate change and water resource shortage is mentioned, but the uniqueness of the study area and its impact on agriculture can be more specifically described. Data or references on the specific water resource situation and agricultural development challenges in the western Nile Delta of Egypt can be added.

Response: The uniqueness of the study area and its impact on agriculture have been described. Moreover, the specific water resource situation and agricultural development challenges in the western Nile Delta of Egypt have been added.

Comment 3. In the methods section, when describing fuzzy logic, a more detailed explanation of how the upper and lower limit values are selected and how they are related to crop demands can be provided.

Response: A detailed description of the methodology adopted to determine lower and upper limits of crop requirements was provided in the materials and methods section (See section 2.3.2.)

Comment 4. The data presentation in the results section needs to be more intuitive. Tables or charts can be used to enhance readability. For example, the distribution ratio of different crop suitability grades can be transformed into bar charts or pie charts.

Response: in Table 7, we provided a detailed description for areas of suitability classes both in hectare (ha) and square kilometers (km2). Besides, the distribution maps (Figures 5 to 8) illustrate the distribution maps of suitability classes.

Comment 5. The discussion section needs to make a more in-depth comparison of the results with existing studies, especially with the research results of other arid regions. In addition, the limitations of the method, such as sample size and spatial resolution, can be discussed, as well as the directions for future research.

Response: Deep comparisons of the results highlighted in our study with the existing national and international studies. Moreover, the limitations of the method, such as sample size and spatial resolution, have been illustrated in the discussion section. The directions for future research have also provided in the discussion section.

Comment 6. The use of professional terms is inconsistent, with "center pivot" and "central pivot" being used interchangeably.

Response: The term “Center pivot” has been correctly to be used in the manuscript.

Response to reviewer 2

Comment 1. What was the selection of criteria or factors based on? Please explain.

Response: The selection of criteria based on available international literature as well as national standards set by the Ministry of Agriculture and Land Reclamation. The references used for selecting these criteria have been added in the revised version (See section 2.3.2.).

Comment 2. In the discussion section, it is better to make comparisons with the research of others and the research done, and for the presented arguments, be sure to use valid and up-to-date references.

Response: Results of the current study were compared with those reported for similar regions under arid and semi-arid environments. The results were also compared with other research studies conducted in the western Nile Delta region. The valid up-to-date references have been added to improve the quality of discussion section.

Comment 3. The advantages and disadvantages of the research done should be said.

Response: The advantages and disadvantages of the current research have been addressed in the discussion section.

Comment 4 Please provide appropriate and valid references for all provided relationships.

Response: Appropriate and valid references for all provided relationships have been added.

Comment 5. In the AHP method, did you use the opinions of relevant experts in the form of a questionnaire to determine the importance of the factors in the pairwise comparison matrix?

Response: Yes, the judgments of ten local soil experts were obtained through questionnaires and this was highlighted in the revised version (See section 2.3.3.)

Comment 6. Please use the papers

Response: The suggested papers have been used and cited in the revised version to improve the quality of the manuscript.

Comment 7. Please give the names of soils according to the WRB system.

Response: The soil names according to the WRB system have been added (See section 2.1)

Comment 8. What was the accuracy of the methods used ? By which criteria are the methods evaluated ?

Response: The accuracy of the proposed methods was checked through calculating the consistency ratio (CR). In the current work, the CR for all the developed matrices was within the acceptable limits (below 0.10). Moreover, to obtain high accuracy, the AHP was implemented twice using both the arithmetic as well as geometric mean algorithms of the experts’ opinions. Consequently, the method demonstrated the lowest CR was considered for further analysis (See section 2.3.3)

Comments 9. Please check the grammar of the whole text with a native speaker and fix the errors.

Response: The grammar of the whole text has been revised in the updated version.

---

## [Decision Letter · Decision Letter 1]

A comprehensive crop suitability assessment under modern irrigation system in arid croplands

PONE-D-25-16781R1

Dear Dr. Shokr,

We’re pleased to inform you that your manuscript has been judged scientifically suitable for publication and will be formally accepted for publication once it meets all outstanding technical requirements.

Kind regards,

Somayeh Soltani-Gerdefaramarzi, Ph. D.

Academic Editor

PLOS ONE

Additional Editor Comments (optional):

Reviewers' comments:

Reviewer's Responses to Questions

**Comments to the Author**

1. If the authors have adequately addressed your comments raised in a previous round of review and you feel that this manuscript is now acceptable for publication, you may indicate that here to bypass the “Comments to the Author” section, enter your conflict of interest statement in the “Confidential to Editor” section, and submit your "Accept" recommendation.

Reviewer #2: All comments have been addressed

2. Is the manuscript technically sound, and do the data support the conclusions?

Reviewer #2: Yes

3. Has the statistical analysis been performed appropriately and rigorously? 

Reviewer #2: Yes

4. Have the authors made all data underlying the findings in their manuscript fully available?

Reviewer #2: Yes

5. Is the manuscript presented in an intelligible fashion and written in standard English?

Reviewer #2: Yes

6. Review Comments to the Author

Reviewer #2: Dear authors

congratulations.

thanks for your corrections and improvement of manuscript

best wishes

7. PLOS authors have the option to publish the peer review history of their article (what does this mean? ). If published, this will include your full peer review and any attached files.

**Do you want your identity to be public for this peer review?** For information about this choice, including consent withdrawal, please see our Privacy Policy .

Reviewer #2:

---

## [Editor Report · Acceptance letter]

PONE-D-25-16781R1

PLOS ONE

Dear Dr. Shokr,

I'm pleased to inform you that your manuscript has been deemed suitable for publication in PLOS ONE. Congratulations! Your manuscript is now being handed over to our production team.

Kind regards,

on behalf of

Dr. Somayeh Soltani-Gerdefaramarzi

Academic Editor

PLOS ONE